# OpenReview forum: "The Steganographic Potentials of Language Models"
_ICLR.cc/2025/Workshop/BuildingTrust — BuildingTrust_

### Official Review · Reviewer_ppKQ · 2025-02-26
**A good workshop paper, but I'm afraid that it might have overly claimed the implication without direct experiments on communiting malicious hidden messages/.**

**Rating:** 6
**Confidence:** 2

**Review:**

It seems like a great paper on this under-explored space, showing the steganographic potential of LMs.
However, one question I have is whether the hidden messages used in the experiment are not mentioned if they are malicious or benign. If they are benign then it’s not clear whether the accuracy would still be this high when the messages are malicious, right? Because Models could potentially reason internally or explicitly not to communicate/ such malicious messages. thus, I think claiming that “The results suggest that if models collude through steganography, effective AI control [9] becomes nearly impossible.” seems a bit overly claimed to me.

Some other issues:
- The template is incorrect.
- Tense inconsistency in writing (sometimes current tense/past tense).

I am happy to raise ratings when these are addressed, but I think overall, this paper can be accepted since it does provide a useful contribution to the field by showing initial results of the steganographic potential of LMs.

---

### Official Review · Reviewer_tr97 · 2025-03-01

**Rating:** 7
**Confidence:** 4

**Review:**

This paper looks into whether language models can hide secret messages in normal text.  Unlike older methods like minimum entropy coupling, this work uses the language model itself to both encode and decode messages.

The "color experiment" experiment is interesting. It shows that language models might create hidden links between words that humans don't notice.  This could mean two language models can secretly talk about colors while appearing to discuss names to anyone watching.  Using reinforcement learning to train models for steganography is a novel approach, and the paper also explores direct steganography and game scenarios.

However, the paper doesn't clearly discuss if fine-tuning for steganography hurts the model's performance on other tasks.  There's a risk that focusing on hidden messages could make the model worse at normal language tasks.

Also, the paper could compare its method more with minimum entropy coupling and similar techniques.  These older methods could also work with strong language models. Looking ahead, LLM agents could be more powerful, and might leverage external tools e.g. minimum entropy coupling related tools for steganographic. Though it's not directly related to contribution of the paper, it worth more discussion on different trade-offs for these two approaches.

---

### Decision · Program_Chairs · 2025-03-05

**Decision:**

Accept

**Comment:**

This paper explores steganographic capabilities in LLMs. The paper investigates how models fine-tuned via reinforcement learning (RL) can develop covert encoding schemes and apply steganography in real-world scenarios. The experiments highlight the challenges of detecting misalignment when models communicate through hidden messages. Overall, while the paper presents valuable initial findings, further discussion on trade-offs, baseline comparisons, and potential real-world implications would strengthen the work. Given the workshop setting, I recommend acceptance.